# The GARP complex is required for cellular sphingolipid homeostasis

Florian Fröhlich[1,2], Constance Petit[1,2], Nora Kory[1,2], Romain Christiano[1,2], Hans-Kristian Hannibal-Bach[3], Morven Graham[4], Xinran Liu[4,5], Christer S Ejsing[3], Robert V Farese Jr[1,2,6]*, Tobias C Walther[1,2,6,7]*

[1]Department of Genetics and Complex Diseases, Harvard T.H. Chan School of Public Health, Boston, United States; [2]Department of Cell Biology, Harvard Medical School, Boston, United States; [3]Department of Biochemistry and Molecular Biology, VILLUM Center for Bioanalytical Sciences, University of Southern Denmark, Odense, Denmark; [4]Center for Cellular and Molecular Imaging, Yale School of Medicine, New Haven, United States; [5]Department of Cell Biology, Yale School of Medicine, New Haven, United States; [6]Broad Institute, Cambridge, United States; [7]Howard Hughes Medical Institute, Harvard T.H. Chan School of Public Health, Boston, United States

**Abstract** Sphingolipids are abundant membrane components and important signaling molecules in eukaryotic cells. Their levels and localization are tightly regulated. However, the mechanisms underlying this regulation remain largely unknown. In this study, we identify the Golgi-associated retrograde protein (GARP) complex, which functions in endosome-to-Golgi retrograde vesicular transport, as a critical player in sphingolipid homeostasis. GARP deficiency leads to accumulation of sphingolipid synthesis intermediates, changes in sterol distribution, and lysosomal dysfunction. A GARP complex mutation analogous to a *VPS53* allele causing progressive cerebello-cerebral atrophy type 2 (PCCA2) in humans exhibits similar, albeit weaker, phenotypes in yeast, providing mechanistic insights into disease pathogenesis. Inhibition of the first step of de novo sphingolipid synthesis is sufficient to mitigate many of the phenotypes of GARP-deficient yeast or mammalian cells. Together, these data show that GARP is essential for cellular sphingolipid homeostasis and suggest a therapeutic strategy for the treatment of PCCA2.

*For correspondence: robert@hsph.harvard.edu (RVF); twalther@hsph.harvard.edu (TCW)

**Competing interests:** The authors declare that no competing interests exist.

## Introduction

Eukaryotic membranes are composed of a complex mixture of lipids belonging to three major classes: sphingolipids, sterols, and glycerophospholipids. The lipid composition of membranes is tightly regulated to achieve homeostasis. Several mechanisms control intracellular levels of glycerophospholipids (*Loewen et al., 2004*; *Martin et al., 2007*) and sterols (*Brown and Goldstein, 1999*; *Sever et al., 2003*; *Hampton and Garza, 2009*). How cells maintain appropriate sphingolipid levels, however, is much less understood.

Sphingolipids make up between 10 and 20% of the mammalian plasma membrane (*van Meer et al., 2008*). They are generated in the ER and Golgi and are subsequently delivered to the plasma membrane by vesicular transport (*Klemm et al., 2009*). Sphingolipids at the plasma membrane are internalized by endocytosis and delivered to the lysosome for catabolism. Sphingolipids can also be recycled from endocytic vesicles back to the plasma membrane via the Golgi apparatus (*Choudhury et al., 2002*). Maintenance of membrane sphingolipid levels is important to ensure plasma membrane integrity and to facilitate normal membrane trafficking at the Golgi apparatus and throughout the endocytic pathway (*Trajkovic et al., 2008*; *Klemm et al., 2009*; *Shen et al., 2014*). In addition,

**eLife digest** Every cell is enveloped by a membrane that forms a barrier between the cell and its environment. This membrane contains fat molecules called 'sphingolipids', which help to maintain the structure of the membrane and enable it to work correctly. These molecules are also used as signals to send information around the interior of the cell and are required for the cell to grow and divide normally. The levels of sphingolipids in the membrane have to be tightly controlled because any imbalance can cause stress to the cell and can lead to serious diseases.

Sphingolipids are made inside the cell and are then sent to a compartment called the Golgi before being delivered to the membrane. To regulate the amount of sphingolipids in the membrane, these molecules are routinely returned to the interior of the cell in small structures called endosomes. From here, they can either be broken down or recycled back to the membrane via the Golgi.

A group of proteins known as the Golgi-associated retrograde protein complex (or GARP) is involved in the movement of endosomes from the membrane to the Golgi. People that have a mutation in the gene that encodes GARP suffer from a severe neurodegenerative disease known as 'progressive cerebello-cerebral atrophy type 2' (PCCA2) in which brain cells die prematurely. Researchers have assumed that the most important role of GARP is to sort proteins, and that the missorting of proteins leads to PCCA2.

Here, Frohlich et al. used a combination of genetic analysis and biochemical techniques to study GARP in yeast cells. The experiments show that GARP is critical for sphingolipid recycling, and that a lack of GARP leads to more sphingolipids being degraded, which results in a build-up of toxic molecules. Frohlich et al. generated yeast cells that have the same mutations in the gene that encodes GARP as those in human patients with PCCA2. These cells grew much slower than normal yeast and were less able to transport sphingolipids from the endosome to the Golgi.

Like the yeast cells, human cells in which the gene that encodes GARP was less active also accumulated toxic molecules. Together, these findings suggest that a build-up of toxic fat molecules may be responsible for the symptoms observed in PCCA2 patients. A future challenge is to find out whether this also applies to patients with Alzheimer's disease and other conditions that also affect endosomes.

intermediates of sphingolipid synthesis, including ceramide and sphingosine-1-phosphate, are important signaling molecules that mediate cell proliferation, differentiation, and death (*Hannun and Obeid, 2008*; *Maceyka and Spiegel, 2014*; *Montefusco et al., 2014*).

How cells achieve sphingolipid homeostasis in membranes remains unclear. However, recent studies have shed some light on this process. In yeast, a reduction in plasma membrane sphingolipid levels results in activation of the Ypk1/2 kinases through the TORC2-signaling cascade (*Roelants et al., 2011*; *Berchtold et al., 2012*). Substrates of these kinases, the ER-localized Orm1/2 proteins, are negative regulators of serine-palmitoyl transferase (SPT), an enzyme that catalyzes the first and rate-limiting step of sphingolipid synthesis (*Breslow et al., 2010*). The phosphorylation of Orm1/2 releases these proteins from SPT, thereby initiating sphingolipid synthesis and the generation of the early synthesis intermediates, long-chain bases. In addition, Ypk1/2 kinases phosphorylate key subunits of the sphingolipid metabolic enzyme ceramide synthase, increasing its activity and flux of more distal steps of sphingolipid biosynthesis (*Muir et al., 2014*).

The physiological importance of sphingolipid homeostasis is underscored by the links of sphingolipid imbalances to disease. For instance, a number of lysosomal storage diseases, such as Niemann–Pick type C or Gaucher's disease, result from lysosomal accumulation of different sphingolipid species (*Platt, 2014*). These diseases preferentially affect the central nervous system (*Platt, 2014*) suggesting that sphingolipid imbalance can lead to neurotoxicity. There are few effective treatments. Therefore, a more complete understanding of sphingolipids is needed and is of significant biomedical importance.

Here, we use a chemical genetics screen to uncover an important role of retrograde endosome-to-Golgi trafficking in maintaining cellular sphingolipid homeostasis. We discover a critical role of the Golgi-associated retrograde protein (GARP) complex in sphingolipid homeostasis, and we provide mechanistic insight into a human mutation in this complex that causes the early-onset neurodegenerative disease

progressive cerebello-cerebral atrophy type 2 (PCCA2) (*Feinstein et al., 2014*). Remarkably, we find that inhibition of sphingolipid synthesis is sufficient to restore crucial aspects of membrane homeostasis when retrograde trafficking via the GARP complex is defective.

## Results

### Endosome-to-Golgi retrograde trafficking is required for sphingolipid homeostasis

To identify genes involved in the regulation of sphingolipid homeostasis, we performed a quantitative, genome-wide screen in yeast for modulators of a growth defect caused by myriocin, an inhibitor of SPT, which catalyzes the first and rate-limiting step of sphingolipid synthesis (*Miyake et al., 1995*). Of 5500 yeast strains analyzed, 703 showed a significant interaction with myriocin (p < 0.05), including 326 that suppressed and 377 that exacerbated the growth defect (*Figure 1A* and *Supplementary file 3*).

One of the strongest class of suppressors identified in the screen (p < $10^{-7}$) contained factors mediating retrograde trafficking from endosomes to the Golgi (*Figure 1—figure supplement 1*). This included mutants in each subunit of the GARP complex (*vps51Δ, vps52Δ, vps53Δ,* and *vps54Δ*) and in four of five subunits of retromer (*vps17Δ, pep8Δ, vps35Δ, vps29Δ*; *Figure 1B*). In addition, our screen identified two of the three SNARE proteins important for GARP-dependent trafficking (*tlg2Δ,* *vti1_{DAMP}*) and *VPS63*, a gene overlapping almost completely with the GTPase *YPT6* that is involved in Golgi-endosomal trafficking. Consistent with a function of Ypt6 maintaining sphingolipid homeostasis, deletion of one subunit of its guanine nucleotide exchange factor, *RIC1*, suppressed the growth defect caused by sphingolipid biosynthesis inhibition. However, *YPT6* had no significant phenotype in our screen. Similarly *IMH1*, encoding a protein important for retrograde transport from endosomes to the Golgi had no phenotype (*Supplementary file 3*). This could indicate that *YPT6* and *IMH1* are false negatives in our screen (e.g., due to problems of library yeast strains) or indicate they are less critical when sphingolipid synthesis is inhibited.

In contrast to phenotypes for genes encoding GARP subunits, the disruption of genes involved in related vesicular trafficking machinery, such as the COG or TRAPP complexes(*Whyte and Munro, 2002*; *Sacher et al., 2008*), resulted in little change in growth when sphingolipid synthesis was impaired by myriocin treatment (*Figure 1—figure supplement 1*; *Supplementary file 4*).

To validate these results, we spotted GARP complex mutants and control strains on plates containing myriocin. The growth defects in yeast cells harboring GARP mutations were suppressed by myriocin, whereas wild-type cell growth remained impaired (*Figure 1C*).

### GARP mutants accumulate upstream intermediates of the sphingolipid synthesis pathway

We hypothesized that the deficiency of the GARP complex may result in the accumulation of a toxic sphingolipid intermediate that is reduced by myriocin treatment. To identify which lipids might contribute to this toxicity, we inhibited key steps of sphingolipid synthesis and examined their effect on cell growth (for an overview see *Figure 2—figure supplement 1*). In contrast to myriocin treatment, the inhibition of downstream steps of sphingolipid synthesis, such as those catalyzed by Aur1, an inositolphosphorylceramide synthase, or ceramide synthase, by using aureobasidin A (*Nagiec et al., 1997*) and fumonisin B1(*Wu et al., 1995*), respectively, strongly inhibited the growth of yeast harboring GARP mutations (*Figure 2A,B*). This suggests that *vps53Δ* cells accumulate a toxic intermediate upstream ceramide synthase and may not have adequate levels of the downstream products.

Consistent with this hypothesis, *vps53Δ* cells but not wild-type cells overexpressing the alkaline ceramidase Ypc1, which is predicted to deplete ceramides and as a consequence downstream sphingolipids showed almost no detectable growth (*Figure 2C*). Also consistent with the hypothesis, *vps53Δ and vps54Δ* cells, but not wild type cells, were highly sensitive to addition of the upstream sphingolipid synthesis intermediate phytosphingosine (PHS) (*Figure 2D*).

To directly assess whether upstream sphingolipid intermediates accumulate in GARP complex-deficient cells, we analyzed cellular lipids by mass spectrometry. Strikingly, *vps53Δ* cells showed an eightfold increase in levels of total long-chain bases compared with wild-type controls (*Figure 2E*). Among the different long-chain base species, dihydrosphingosine (DHS) increased ~tenfold and PHS increased ~threefold (*Figure 2F,G*). In addition, *vps53Δ* cells had a ~twofold reduction of the complex sphingolipid M(IP)$_2$C (*Figure 2E*); the levels of IPC, MIPC, and ceramides were unchanged.

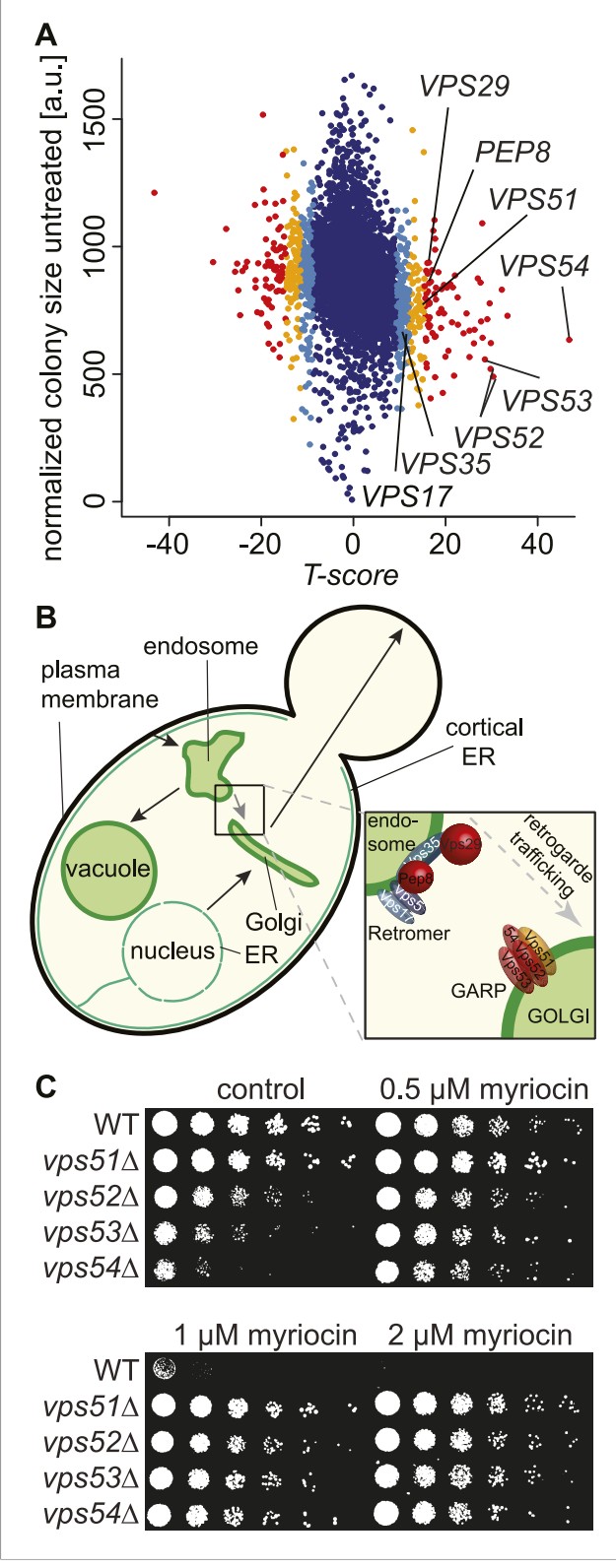

**Figure 1**. A chemical biology screen reveals the retrograde endosome to Golgi trafficking machinery as a key regulator of sphingolipid homeostasis. (**A**) A chemical genetic screen for interactions with myriocin. Calculated T-scores are plotted against colony size on control plates. Genes are color-coded according to their significance score. Red p < 0.001, orange p < 0.01, light blue p < 0.05, blue p > 0.05. (**B**) A model for retromer- and

*Figure 1. Continued*

Golgi-associated retrograde protein (GARP)-mediated retrograde endosome-to-Golgi trafficking. Subunits of the GARP complex and retromer identified in our screen are denoted. Genes are color coded according to their T-score. (**C**) Depletion of sphingolipids suppresses the growth defect of yeast strains harboring GARP mutations. Wild-type cells and *vps51Δ*, *vps52Δ*, *vps53Δ*, and *vps54Δ* mutant cells were spotted on control plates and plates containing increasing concentrations of myriocin, as indicated.

The following figure supplement is available for figure 1:

**Figure supplement 1**. GO analysis of all suppressing mutants from the chemical genomic myriocin screen.

Based on the current model of sphingolipid synthesis regulation, we expect that reduction of the complex sphingolipid $M(IP)_2C$ in *vps53Δ* cells may activate the upstream sphingolipid synthesis enzyme SPT by releasing Orm1/2 inhibition, thereby exacerbating the GARP mutant phenotype. Consistent with this model, we found that Orm1 was hyperphosphorylated in *vps53Δ* cells compared with controls (*Figure 2H*).

Together, these results suggest that blocking the GARP-mediated recycling of lipids to the plasma membrane leads to a toxic build-up of long-chain bases, possibly due to a higher rate of degradation of complex sphingolipids in the vacuole and increased biosynthesis. We therefore reasoned that myriocin treatment reduces this toxicity by lowering the levels of long-chain bases. To evaluate this possibility, we determined the levels of long-chain bases before and after myriocin treatment in *vps52Δ*, *vps53Δ*, *vps54Δ*, and control cells. Importantly, myriocin treatment greatly reduced the accumulation of long-chain bases in each of the GARP mutants (*Figure 2F,G*). The rate of long-chain base decrease during the myriocin treatment time course was similar in wild-type and *vps53Δ* cells (*Figure 2—figure supplement 2*), arguing that myriocin is equally effective in reducing sphingolipid synthesis in either strain. However, even after prolonged, DHS levels remained elevated compared with untreated control cells, suggesting the pool of long-chain bases turns over more slowly (*Figure 2F*).

## Long-chain base accumulation leads to altered vacuolar morphology and function in GARP mutants

We reasoned that complex sphingolipids fail to be recycled to the plasma membrane in GARP mutants and are instead rerouted for degradation in vacuoles causing accumulation of long-chain bases and triggering lipotoxicity. A prediction from this hypothesis is that *vps53Δ* and wild-type cells would distribute exogenously added, fluorescently labeled sphingosine (NBD-sphingosine) differently. Testing this possibility, we found that added NBD-sphingosine and FM4-64, a marker of endocytic membranes both initially label the plasma membrane, but then segregate into different compartments in wild-type cells: as expected, after 60 min, FM4-64 stained the yeast vacuole, whereas the NBD-sphingosine signal localized in one or a few foci likely representing compartments of the endosomal/secretory pathway (*Figure 3A*, top control panels). In *vps53Δ* cells, however, both lipids co-localized after 60 min in what appeared to be highly fragmented vacuoles (*Conboy and Cyert, 2000*; *Conibear and Stevens, 2000*) (*Figure 3A* middle control panels). Intriguingly, the abnormal vacuolar morphology in *vps53Δ* cells was partially rescued by 12-hr myriocin treatment, resulting in a few small vacuoles (*Figure 3A* bottom-myriocin, *3B, 3C*, and *Figure 3—figure supplement 1* for characterization of vacuolar classes). However, NBD-sphingosine still localized to the vacuoles in myriocin-treated *vps53Δ* cells, not to the plasma membrane as in wild-type cells (*Figure 3A*). Together, these results suggest that exogenously added long-chain bases are maintained to a large degree in the endosomal/secretory pathway of wild-type cells, but accumulate in the vacuole of GARP mutants. The data also suggest that myriocin treatment of these cells does not restore proper endosome to Golgi recycling of sphingolipids, but partially rescues vacuolar dysfunction, as indicated by restoration of vacuole morphology.

To further understand the physiological impact of trafficking defects caused by GARP complex deficiency, we analyzed the protein composition of *vps53Δ* mutant cells by mass spectrometry-based proteomics. Among the 3347 proteins analyzed, we found that the levels of three proteins under control of the zinc-dependent transcription factor Zap1 (*Lyons et al., 2000*) were strongly decreased

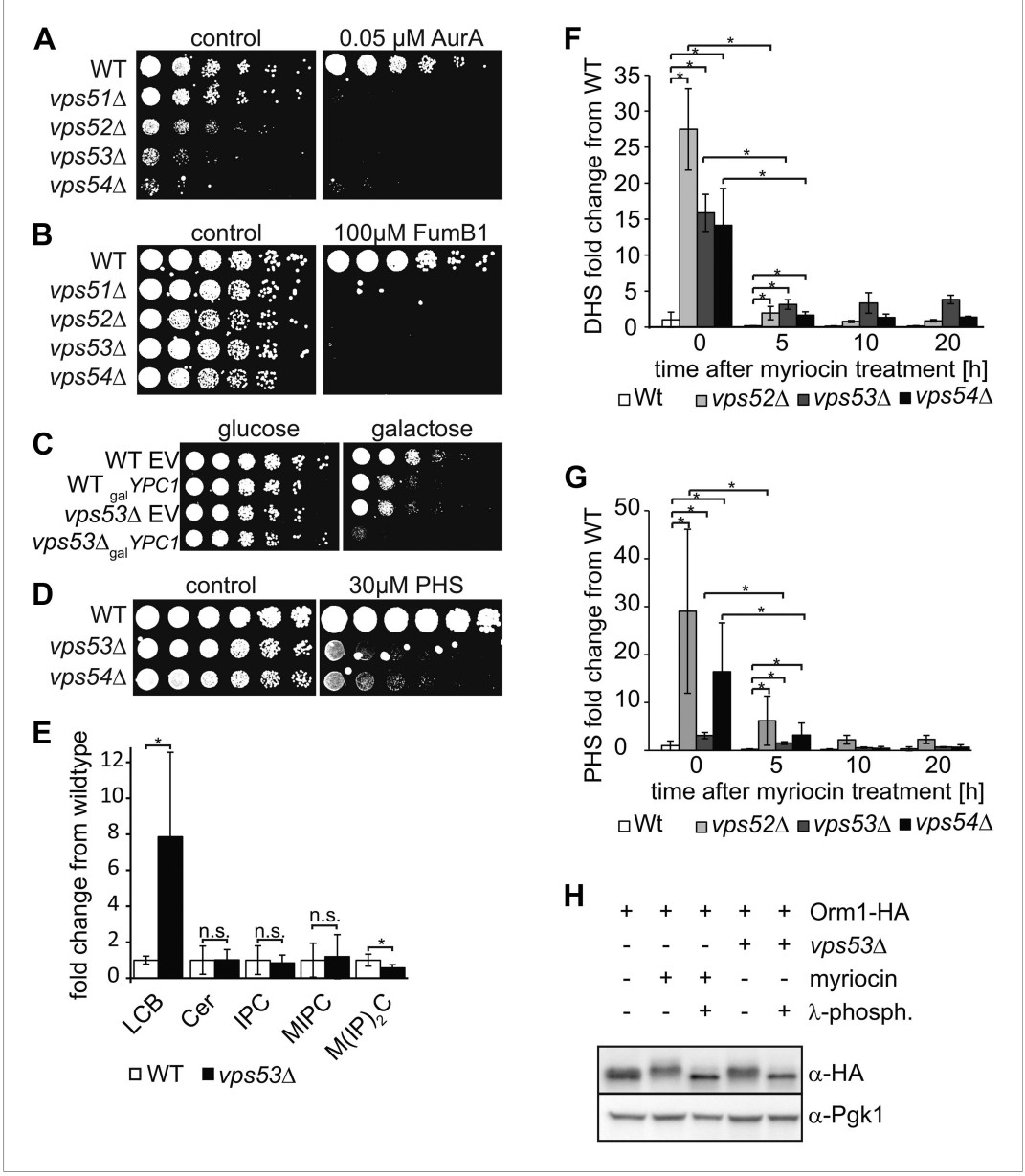

**Figure 2**. The disruption of the GARP complex leads to the accumulation of early sphingolipid synthesis intermediates. (**A**, **B**, **C**) Blocking early steps of sphingolipid synthesis exacerbates GARP-associated growth defects. (**A**) GARP mutants are sensitive to IPC synthase inhibition. Wild-type, *vps51Δ*, *vps52Δ*, *vps53Δ*, and *vps54Δ* cells were spotted on control plates and plates containing 0.05 μM aureobasidin A. (**B**) GARP mutants are sensitive to ceramide synthase inhibition. Wild-type, *vps51Δ*, *vps52Δ*, *vps53Δ*, and *vps54Δ* cells were spotted on control plates and plates containing 100 μM fumonisin B1. (**C**) *VPS53* mutants are sensitive to overexpression of the alkaline ceramidase Ypc1. Wild-type or *vps53Δ* cells harboring an empty plasmid or a plasmid encoding YPC1 under control of the *GAL10* promoter were spotted on glucose- or galactose-containing plates. (**D**) GARP mutants are sensitive to high levels of long-chain bases, early sphingolipid intermediates. Wild-type, *vps53Δ*, and *vps54Δ* cells were spotted on control plates or plates containing 30 μM phytosphingosine (PHS). (**E**) GARP complex deficiency results in an accumulation of long-chain bases. The lipidomic analysis of sphingolipids from *vps53Δ* (black) compared with wild-type strains (white) is shown. LCB = long chain base; CER = ceramide; IPC = inositolphosphorylceramide; MIPC = mannosylinositolphosphorylceramide; M(IP)₂C = mannose-(inositol-P)2-ceramide. *$p < 0.05$; n.s. not significant (**F**, **G**) Long-chain bases in GARP mutants are reduced upon myriocin treatment but remain elevated. The levels of (**F**) dihydrosphingosine (DHS) and (**G**) PHS from wild-type (white bars), vps52Δ (light gray bars), vps53Δ (dark gray bars), and vps54Δ cells (black bars) to myriocin treatment is plotted as fold change from wild-type. *$p < 0.05$; n.s. not significant (**H**) Orm1/2 proteins are hyperphosphorylated in *vps53Δ* mutants. Orm1-HA expressing wild-type or

*Figure 2. Continued*

vps53Δ cells were analyzed by Western blotting against the HA tag or PGK1 as control. Wild-type cells were treated with 5 µM myriocin as indicated. Treatment of the cell lysates with λ-phosphatase as indicated.

The following figure supplements are available for figure 2:

**Figure supplement 1**. Model of Sphingolipid metabolism.

**Figure supplement 2**. The rate of serine palimtoyl-transferase inhibition is similar in WT and vps53Δ cells.

(Adh4, Zrt1, Zps1; $P < 1e^{-11}$; *Figure 3D*). Since the yeast vacuole is the main storage organelle for intracellular zinc (*Simm et al., 2007*), we hypothesized that the altered vacuolar morphology of vps53Δ mutants results in increased zinc release from the vacuole contributing to toxicity. Consistent with previous studies (*Banuelos et al., 2010*), we found that increased levels of zinc exacerbated the growth defect of vps53Δ mutants. Importantly, this sensitivity is completely suppressed by myriocin treatment (*Figure 3E*).

Our proteomic studies also detected significantly ($p < 0.05$) decreased levels of several ER-resident, early sphingolipid metabolic enzymes, such as Tsc10, Sur2, Sur4, or Phs1, potentially indicating regulatory adaptations to high levels of intracellular long-chain bases (*Figure 3D*).

## GARP complex deficiency results in altered ergosterol distribution and lipid droplet accumulation

Sphingolipids and sterols interact in the plasma membrane and are both internalized by endocytosis, suggesting that their levels might be coordinately regulated (*Simons and Vaz, 2004*; *Jacquier and Schneiter, 2012*). We therefore reasoned that sphingolipid imbalance due to GARP deficiency could also lead to sterol accumulation or altered cellular distribution. Indeed, filipin, a dye that binds sterols, accumulated in internal structures of vps53Δ and vps54Δ cells (*Figure 4A*). Inhibition of sphingolipid synthesis by myriocin in GARP mutants was sufficient to reverse sterol accumulation, as detected by filipin staining. In contrast, wild-type cells accumulated filipin-positive structures during myriocin treatment. This finding is consistent with the sterol accumulation phenotype of the temperature sensitive SPT mutant lcb1-100 (*Baumann et al., 2005*).

Lipidomic analyses showed no significant change in cellular ergosterol levels in vps53Δ cells. However, we detected a two to threefold increase in ergosterol esters in these cells, suggesting that neutral lipids accumulate due to GARP complex deficiency (*Figure 4B*). Consistent with this, vps53Δ mutants showed a threefold increase in lipid droplets per cell, as identified by the LD marker protein Faa4, compared with wild-type cells (*Figure 4C,D*).

To further explore the role of GARP in maintaining sterol homeostasis, we analyzed genetic interaction data (*Hoppins et al., 2011*) of vps52Δ and found synthetic growth defects with mutations in several genes encoding enzymes of ergosterol biosynthesis (*ERG2, ERG20, ERG25, ERG26*) (*Figure 4E*). Tetrad analyses of vps53Δ erg3Δ double mutants confirmed these synthetic growth defects (*Figure 4—figure supplement 1*). In addition, *VPS52* showed positive correlations with many genes in ergosterol metabolism (*ERG3, ERG25, ERG26, ERG6, ERG2*, and *ERG4*; *Figure 4F*), further suggesting a role in maintaining intracellular sterol levels. Together, these results suggest that the GARP complex may help manage sterol levels in cells by recycling them from endosomes to the plasma membrane via the Golgi apparatus.

## A human *VPS53* mutation causing neurodegeneration is a partial loss-of-function mutant in yeast assays

Compound heterozygous mutations of human *VPS53* were reported recently to cause the neurodegenerative disease PCCA2 (*Feinstein et al., 2014*). In these patients, one Vps53 allele abolishes protein expression, and the second is a missense mutation (Q695R) in the highly conserved C-terminus of VPS53. To understand the molecular consequences of the latter mutation, we generated a yeast strain harboring an analogous mutation, *Q624R*, and investigated its effects on lipid homeostasis, retrograde trafficking of proteins between endosomes and the Golgi, and vacuolar

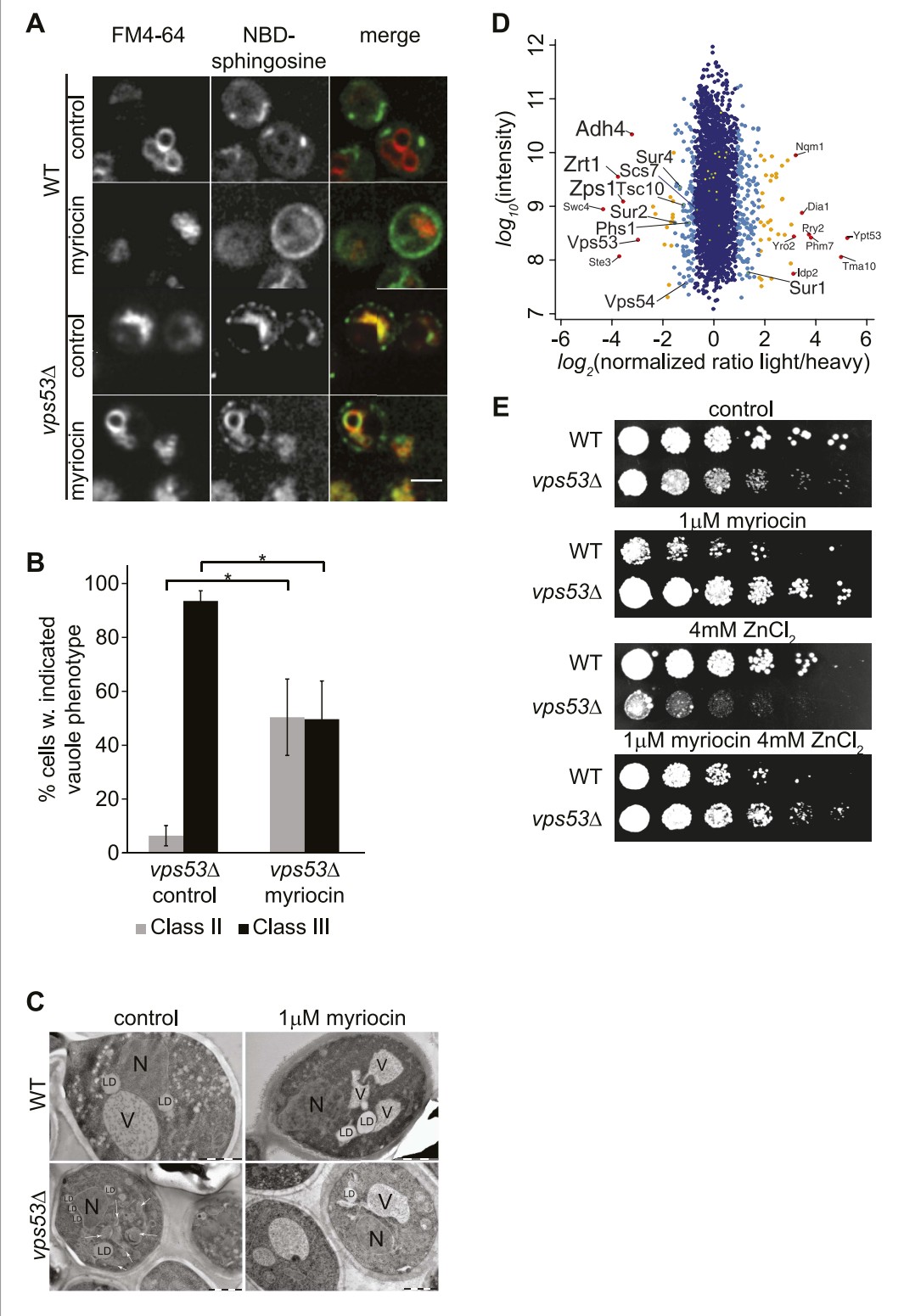

**Figure 3**. GARP mutants show altered vacuolar morphology and function. (**A**) Sphingolipid recycling remains blocked upon myriocin treatment and sphingolipids accumulate in vacuoles. Wild-type cells (top panels), wild-type cells treated with myriocin (1 μM, 12hr), vps53Δ cells, or vps53Δ cells treated with myriocin (1 μM, 12hr). NBD-sphingosine, green. Vacuoles (FM4-64), red. Representative images are shown. Scale bar = 2.5 μm. (**B**, **C**) Inhibition of sphingolipid biosynthesis partially restores vacuolar morphology in GARP mutants. (**B**) Quantification of the
*Figure 3. continued on next page*

*Figure 3. Continued*

vacuolar phenotypes in *vps53Δ* mutants, mock treated or treated with myriocin is shown (Classification, see Supplementary figure 2). (**C**) Thin-section EM analysis of high-pressure frozen wild-type or *vps53Δ* cells mock-treated (left panels) or treated with myriocin (12hr, 1 µM; right panels). Representative images are shown. N = nucleus; V = vacuole; LD = lipid droplet. White arrows indicate fragmented vacuoles. Scale bars = 1 µM (**D**). Three targets of zinc transcription factor Zap1 are down regulated in the GARP mutant *vps53Δ*. A proteomic analysis of *vps53Δ* and wild-type cells is shown. Protein intensities are plotted against light/heavy SILAC ratios. Significant outliers are colored in red ($P < 1^{-11}$), orange ($P < 1^{-4}$), or light blue (p < 0.05); other proteins are shown in dark blue. **E**) GARP mutant zinc sensitivity is rescued by myriocin treatment. Wild-type and *vps53Δ* cells were spotted on control plates or plates containing 1 µM myriocin, 4 mM zinc or 1 µM myriocin and 4 mM zinc.

The following figure supplement is available for figure 3:

**Figure supplement 1**. Classification of vacuolar phenotypes used for quantification.

morphology. Similar to *vps53Δ* cells, cells harboring the *Q624R* mutation were resistant to myriocin treatment (**Figure 5A**). However, growth in the mutant was more impaired than in *vps53Δ* cells, suggesting a partial loss of GARP function in *vps53 Q624R* cells. Subsequent proteomic analyses revealed that the *vps53 Q624R* mutant protein is expressed at similar levels as the wild-type protein, indicating that this effect is due to its impaired activity, not its instability (**Figure 5—figure supplement 1**).

To evaluate whether endosome-to-Golgi trafficking is also compromised in *vps53 Q624R* cells, we evaluated this process by assaying the localization of Vps10, the sorting receptor for the vacuolar protease carboxypeptidase Y (**Chi et al., 2014**). In this assay, a green fluorescent protein (GFP)-tagged form of Vps10 is retrieved from endosomes to the Golgi apparatus by retrograde trafficking (**Conibear and Stevens, 2000**). We quantified co-localization of Vps10-GFP with the Golgi apparatus (marked with Sec7-*tomato*) or the endosome (marked by Vps17-*tomato*) (**Figure 5B**). Using this assay, we found that cells harboring a *vps53 Q624R* mutant allele had significantly reduced Vps10 levels in the Golgi apparatus (28%, compared with 43% in wild-type cells) and increased signal in the endosome (49%, compared with 24% in wild-type cells; **Figure 5C**), suggesting that retrograde trafficking is indeed impaired.

Also consistent with a partial loss of GARP function, *vps53 Q624R* mutant cells showed vacuolar fragmentation, highlighted by multiple small vacuoles in cells, which we found three times more frequently than in control cells (33% vs 11%, **Figure 5D,E**, **Figure 3—figure supplement 1** for characterization of vacuolar classes). This intermediate phenotype appears to be similar to vacuoles in *vps53Δ* cells depleted for sphingolipids (**Figure 3A**) and is much weaker than in *vps53Δ* mutants in which all cells have highly fragmented vacuoles with no typical round structures apparent (**Figure 5D,E**).

## GARP deficiency-associated defects in mammalian cells are suppressed by inhibition of sphingolipid synthesis

To determine whether GARP's function in sphingolipid homeostasis is conserved in mammals, we assessed the phenotype of HeLa cells depleted of the GARP complex. Similar to our findings in yeast, knock-down of the *VPS53* GARP subunit caused accumulation of cholesterol in internal structures of HeLa cells, based on filipin staining (**Figure 6A** and (**Perez-Victoria et al., 2010**)). Notably, this accumulation was reduced dramatically by myriocin treatment, restoring filipin staining to levels similar to control cells treated with myriocin (**Figure 6B**).

GARP complex deficiency results in a defect in retrograde transport, leading to defects in lysosome morphology in HeLa cells (**Perez-Victoria et al., 2008**). We therefore reasoned that lysosome dysfunction could contribute to the toxicity caused by GARP deficiency. We confirmed that lysosomes appeared swollen and clustered in the juxtanuclear area of Vps53 knock-down cells, although LAMP1 protein levels were normal (**Figure 6C**, **Figure 6—figure supplement 2**). Importantly, myriocin treatment resulted in a more uniform lysosome distribution and partially reduced the size of LAMP-1 positive lysosomes in Vps53 knock-down cells (**Figure 6C**). In contrast, lysosome morphology in control cells was not appreciably affected.

To test whether these defects could be caused by accumulation of sphingolipid metabolism intermediates, we isolated sphingolipids from myriocin- or mock-treated *VPS53* knock-down and

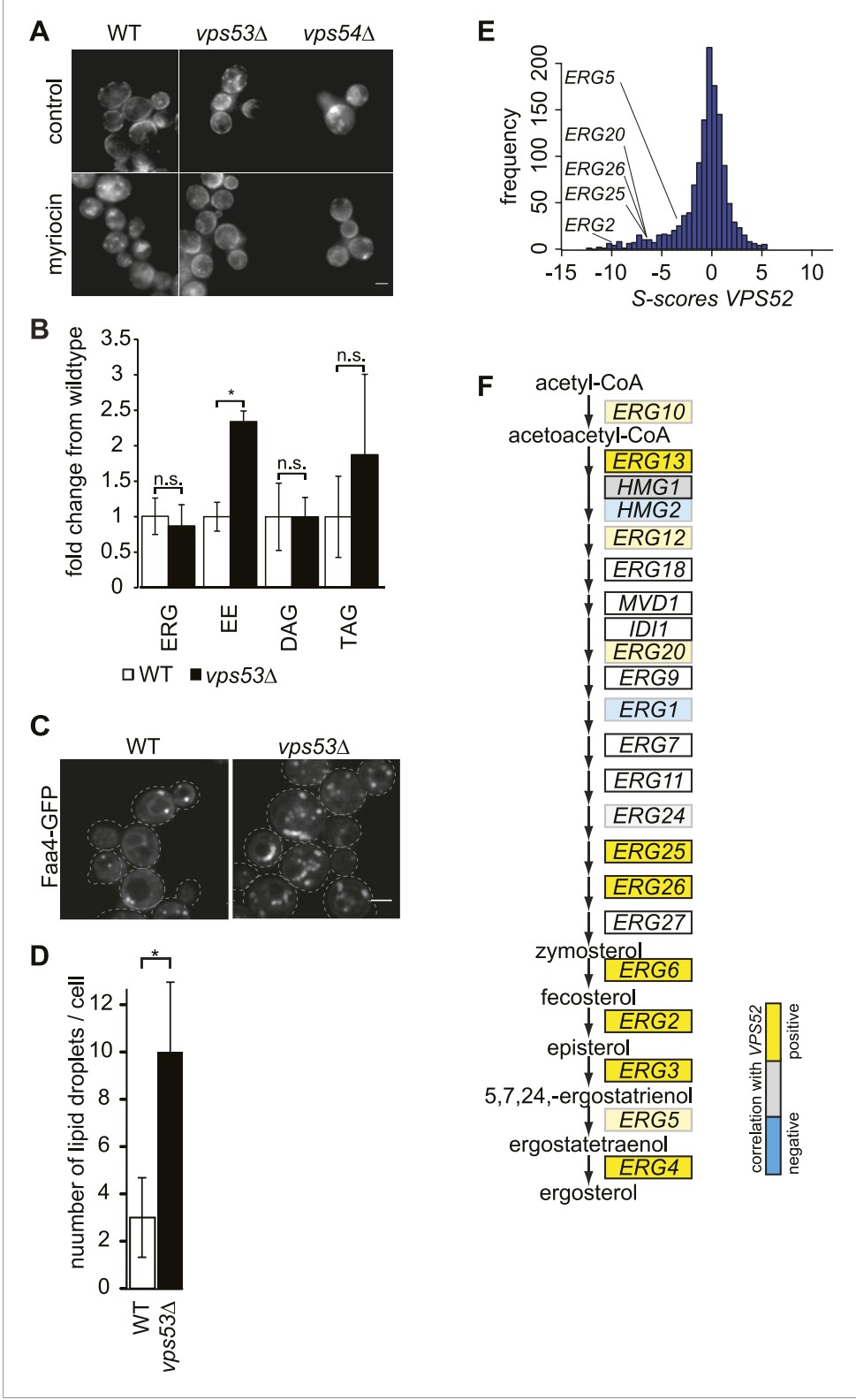

**Figure 4**. The disruption of the GARP complex alters sterol distribution in yeast. (**A**) Intracellular sterols accumulate in the GARP mutants *vps53Δ* and *vps54Δ*. Wild-type (left panels), *vps53Δ* cells (middle panels), and *vps54Δ* cell (right panels) treated with methanol (top panels) or myriocin (lower panels) were stained with filipin and analyzed by epifluorescence microscopy. Representative images are shown. Scale bar = 2.5 μm. (**B**) Neutral lipids accumulate in

*Figure 4. continued on next page*

*Figure 4. Continued*

the GARP mutant *vps53Δ*. Lipidomic analysis of neutral lipids isolated from *vps53Δ* (black) expressed in fold change from the wild-type (white). ERG = ergosterol; EE = ergosterol ester; DAG = diacylglycerol, TAG = triacylglycerol. *p < 0.005; n.s. not significant. (**C, D**) Lipid droplets accumulate in the GARP mutant *vps53Δ*. Lipid droplets marked by Faa4-GFP. (**C**) Representative confocal midsections of wild-type (left) or *vps53Δ* cells are shown. Scale bar = 2.5 μm. (**D**) Quantification of (**C**). The number of lipid droplets per cell compared to wild-type cells (white bar) is shown. n = 100 cells. *p < 0.001; n.s. not significant (**E**) Components of the GARP complex genetically interact with sterol synthesis genes. Histogram of *S*-scores for *VPS52* extracted from an EMAP (*Hoppins et al., 2011*) is shown. (**F**) Model of ergosterol metabolism. Genes are color-coded according to correlation coefficient with the GARP subunit *VPS52*. Positive-correlated genes are indicated in yellow; anti-correlated genes are indicated in blue.

The following figure supplement is available for figure 4:

**Figure supplement 1**. Components of the GARP complex genetically interact with ergosterol metabolism genes in yeast.

control cells and analyzed them by lipidomics. We found a ∼2.5 fold increase in hexosylceramide and a significant increase in the levels of sphingosine and sphinganine, as well as a decrease in sphingomyelin (*Figure 6D* and *Figure 6—figure supplement 1*). Importantly, myriocin treatment restored the levels of these sphingolipid intermediates in *VPS53 knock-down* cells (*Figure 6D*).

To further evaluate the possibility that sphingolipid imbalances could, at least partially, contribute to development of PCCA2, we analyzed sphingolipid levels in fibroblasts from PCCA2 patients and controls. Similar to *VPS53* knock-down in HeLa cells, PCCA2 fibroblasts exhibited increases in sphingosine, sphinganine, and ceramides compared with control fibroblasts. Similar to what was observed for *VPS53* knock-down in HeLa cells, myriocin treatment reduced the accumulation of sphingolipid intermediates in PCCA2 fibroblasts to levels similar to those found in untreated control fibroblasts (*Figure 6E*).

## Discussion

Here, we show that GARP complex-mediated retrograde trafficking from endosomes to the Golgi is important for cellular sphingolipid homeostasis. In yeast, loss of the GARP complex results in reduced levels of complex sphingolipids and the accumulation of sphingolipid synthesis intermediates, most notably long-chain bases. Importantly, this accumulation is correlated with abnormal vacuolar morphology and function, suggesting this build-up may be toxic. The depletion of a GARP complex subunit in mammalian cells results in similar phenotypes. Retrograde trafficking likely influences the subcellular localization of a large number of proteins. It is thus remarkable that many of phenotypes associated with GARP deficiency are greatly attenuated by pharmacological inhibition of SPT, the first and rate-limiting step in sphingolipid synthesis. Together, these results suggest that accumulation of toxic lipids may underlie a substantial degree of cellular dysfunction due to genetic defects in retrograde trafficking and that restoration of sphingolipid balance may be a strategy to treat diseases due to these defects.

Our data suggest that the GARP complex is critical for recycling lipids between the plasma membrane, endosomes, and the Golgi apparatus. In both yeast and mammalian cells, GARP complex deficiency disrupted sphingolipid and sterol homeostasis and resulted in the accumulation of sterols in the endo-lysosomal system. Normally, lipids retrieved from the plasma membrane by endocytosis can be recycled via the retrograde endosome-to-Golgi pathway to supply lipids necessary for rapid membrane expansion during growth. In GARP mutants, however, some membrane-derived lipids appear to be rerouted to vacuoles (or lysosomes) for degradation. This rerouting of lipids likely reduces their levels in the plasma membrane and increases the levels of sphingolipids, and their break-down products, in the vacuole/lysosome. This reduction in membrane sphingolipids may explain GARP's apparent role in plasma membrane organization, reported previously in genome-wide visual screens (*Grossmann et al., 2008*; *Frohlich et al., 2009*). It is yet unclear whether sphingolipids and sterols follow bulk flow of membrane materials in the endocytic pathway or are preferably recycled by retromer and GARP complexes. Regardless, our data are most consistent with the hypothesis that in the absence of GARP, sphingolipids and sterols are rerouted to vacuoles/

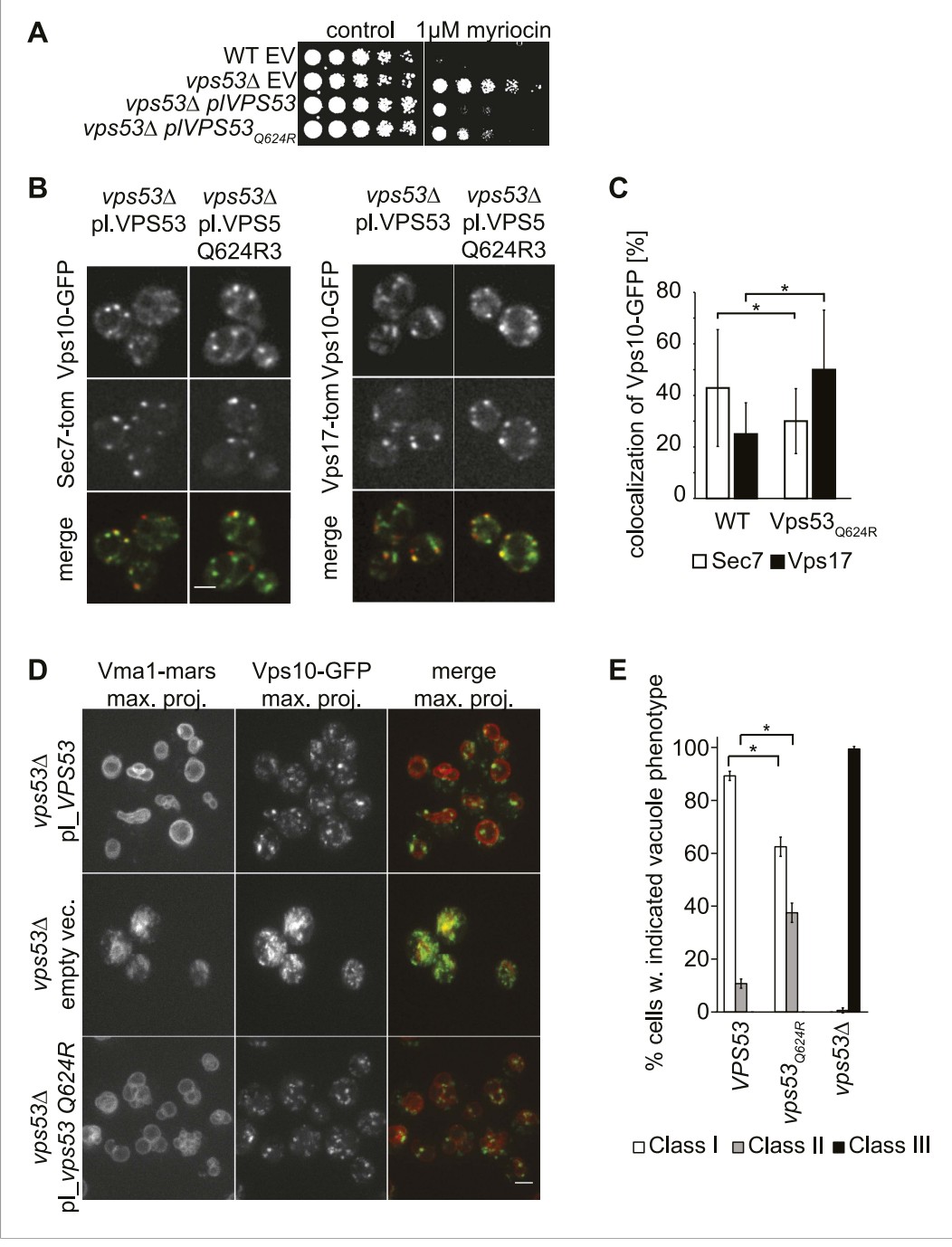

**Figure 5**. A PCCA2-causing GARP complex mutation is a partial loss of function allele. (**A**) A mutation analogous to the *VPS53* allele causing progressive cerebello-cerebral atrophy type 2 (PCCA2) in humans is partially resistant to sphingolipid biosynthesis inhibition induced by myriocin. Wild-type cells harboring an empty plasmid and *vps53Δ* cells harboring an empty plasmid, a plasmid expressing Vps53, or a plasmid expressing the mutant *vps53 Q624R* were spotted on myriocin-containing (right) plates or control plates containing methanol (left) (**B**) Endosome-Golgi trafficking is partially impaired in yeast cells expressing the analogous PCCA2-causing mutation *vps53 Q624R*. The GFP-tagged CPY receptor Vps10-GFP (top panels left and right) was co-expressed with either Sec7-tomato (Sec7-tom; middle left panels) or Vps17-tomato (Vps17-tom; middle right panels) in wild-type and *vps53 Q624R* mutant cells. Representative confocal midsections are shown; scale bar, 2.5 µm. (**C**) The CPY receptor Vps10 accumulates in *vps53 Q624R* endosomes. Quantification of the distribution of Vps10-GFP between the Sec7-decorated Golgi (white bars) and retromer-decorated endosomes (black bars) in wild-type and *vps53 Q624R* cells. *p < 0.005; n.s. not significant. (**D**) Mutations in the GARP complex cause vacuolar fragmentation. Maximum
*Figure 5. continued on next page*

*Figure 5. Continued*

projections of vacuoles marked with RFPmars-tagged V-ATPase subunit Vma1 (left panels) and the GFP-tagged CPY sorting receptor Vps10 (middle panels) in *vps53Δ* cells harboring a plasmid expressing Vps53 (top panels), an empty plasmid (middle panels) or a plasmid expressing a vps53 Q624R mutant (lower panels) are shown; scale bar = 2.5 µm. (**E**) Quantification of (**D**). Cells with class I (white bars), class II (gray bars), and class III (black bars) vacuoles were counted and plotted as percentage of the total number. n = 50. *p < 0.005; n.s. not significant. For phenotypic classification, see *Figure 3—figure supplement 1*.

The following figure supplement is available for figure 5:

**Figure supplement 1**. A GARP complex mutation analogous to the VPS53 allele causing PCCA2 vps53 Q624R does not impact protein stability.

lysosomes. In lysosomes, sphingolipids are broken down by acid sphingomyelinase and ceramidase, likely contributing to the elevation in sphingoid bases. However, it is also possible that missorting of sphingolipid synthesis enzymes or trafficking proteins in GARP mutants contributes to the observed phenotypes. For instance, a protein important for the export of sphingolipid intermediates from lysosomes/vacuoles could be missorted in GARP mutants, thus increasing vacuolar sphingolipid accumulation.

Several of our findings suggest that the accumulation of early sphingolipid synthesis intermediates due to GARP deficiency may result in cellular toxicity. We show that inhibition of SPT, which lowers long-chain bases in GARP mutants, suppresses the growth defect of these strains. In contrast, the inhibition of later steps of sphingolipid synthesis, for example, by blocking ceramide synthase or inositolphosphoryl ceramide synthase, does not lower long-chain base levels and increases toxicity in GARP mutants. In addition, treatment with the long-chain base PHS exacerbated this toxicity. In yeast, low levels of plasma membrane sphingolipids activate TORC2/Slm/Ypk signaling to phosphorylate the Orm-proteins, relieving their inhibition of SPT, increasing sphingolipid synthesis (*Breslow et al., 2010*; *Han et al., 2010*; *Roelants et al., 2011*; *Berchtold et al., 2012*). Consistent with a decrease of plasma membrane sphingolipids, we found that the Orm1/2 proteins are hyperphosphorylated in GARP mutants suggesting that de novo sphingolipid synthesis is also upregulated. This increase may exacerbate GARP deficiency-induced lipotoxicity due to increased production of these early synthesis intermediates.

Why the accumulation of long-chain bases is toxic is currently unclear. One possibility is that long-chain bases possess detergent-like properties, especially in an acidic environment (*Jimenez-Rojo et al., 2014*), such as in the yeast vacuole. In model membranes, such as giant unilamellar vesicles, this detergent-like action leads to vesicular leakage (*Contreras et al., 2006*). This could explain the findings of marked vacuolar fragmentation and potential leakage of zinc from GARP-deficient yeast vacuoles. In support of this hypothesis, we found that inhibition of SPT and reduction of long-chain bases suppressed the zinc sensitivity in GARP mutants in yeast. Also consistent with this hypothesis, the accumulation of sphingolipid intermediates impairs ion homeostasis in mammalian lysosomes of a cellular Niemann–Pick type-C disease model (*Lloyd-Evans et al., 2008*). Alternatively, long-chain bases are known to inhibit glycerolipid synthesis (*Wu et al., 1993*), which might contribute to their cytotoxicity.

Since sphingolipid and sterol levels are coordinated in membranes, it is not surprising that we also found that GARP deficiency alters sterol metabolism. Sphingolipids and sterols are thought to interact with each other in membranes. Thus, GARP deficiency likely results in missorting of both to the vacuole/lysosome. Our data are most consistent with a model where sphingolipids are degraded in the lysosome/vacuole leading to a toxic build-up of sphingolipid metabolites, whereas sterols are exported and stored in cytoplasmic lipid droplets as more inert fatty acid esters. Consistent with this, lowering sphingolipid levels by myriocin treatment of wild-type cells increases sterols stained by filipin. Possibly, this indicates that in the absence of sufficient sphingolipids, sterols that are not complexed with sphingolipids build-up and are stored in ester form. A surprising finding is that the accumulation of sterols in GARP mutants is suppressed by inhibiting sphingolipid synthesis. These data argue that in addition to sterol missorting in GARP mutants, accumulation of sphingolipid intermediates impairs normal sterol homeostasis, for example, by interfering with sterol export from the vacuole/lysosome or synthesis regulation. This further highlights that a primary cause for defects due to GARP deficiency may be lysosomal/vacuolar dysfunction due to sphingolipid accumulation.

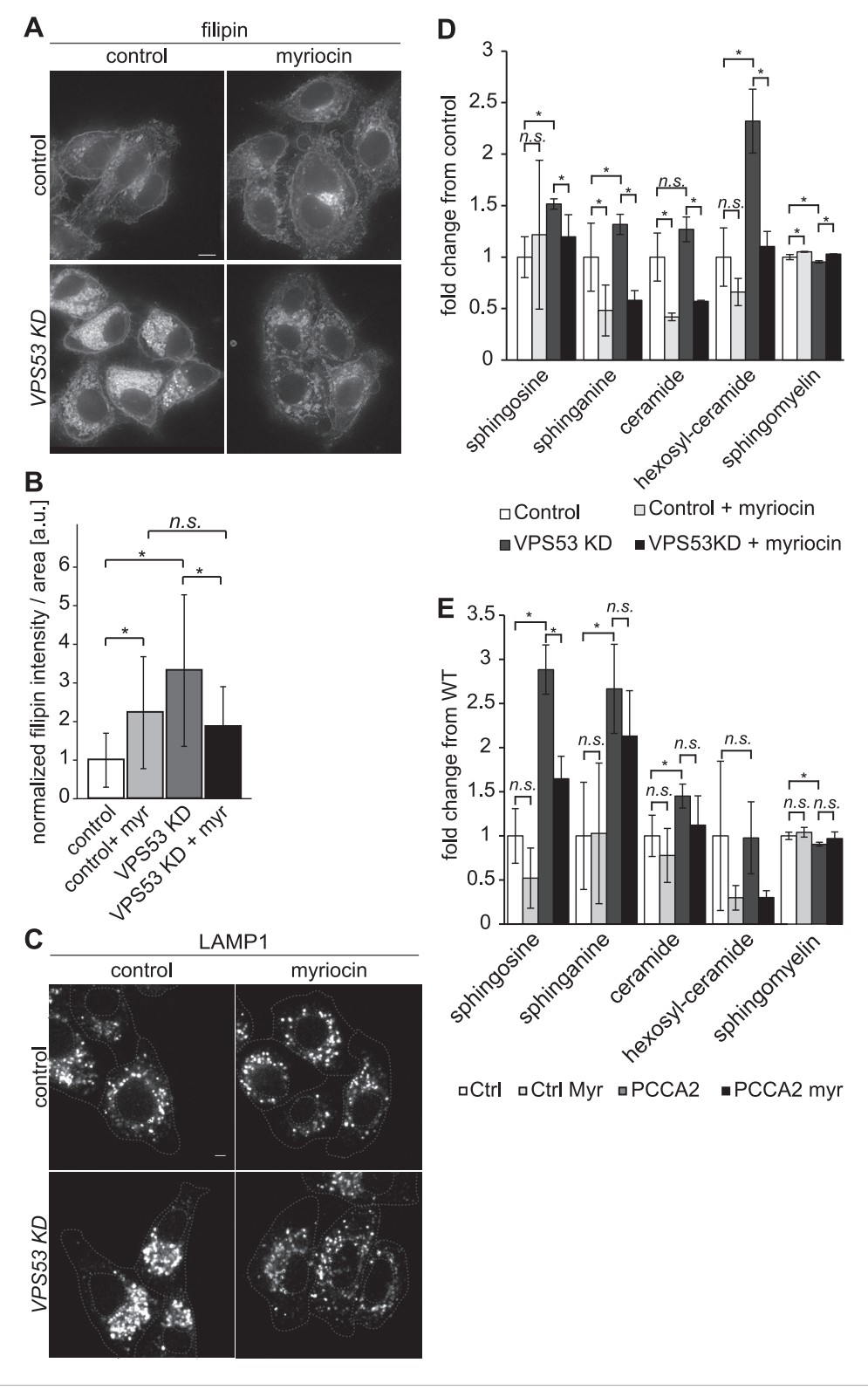

**Figure 6**. The depletion of sphingolipid levels reduced lysosome clustering and sterol accumulation due to GARP complex deficiency in HeLa cells. (**A**, **B**) Myriocin treatment reduced build-up of free cholesterol due to GARP complex deficiency. (**A**) Filipin staining of unesterified cholesterol in control (top panels) or Vps53 knock-down (lower panels) in HeLa cells treated with 1 μM myriocin for 12 hr (right panels) or DMSO as a control (left panels). Representative images are shown. Scale bar = 5 μm. (**B**) Quantification of the average free cholesterol filipin

*Figure 6. continued on next page*

*Figure 6. Continued*

intensity/cell normalized to control cells. n = 32. *p < 0.0005; n.s. not significant. (**C**) Myriocin treatment partially restored intracellular distribution of lysosomes in the GARP KD. Control cells (top panels) or Vps53-KD cells (lower panels) were treated with 1 μM myriocin for 12 hr (right panels) or DMSO as a control (left panels) and stained with an antibody against the lysosomal protein LAMP-1. Representative confocal midsections are shown; scale bar = 2.5 μm. (**D**) Myriocin treatment reduced accumulations of early sphingolipid intermediates due to GARP complex deficiency. Lipidomic analysis of mock-treated HeLa control cells (white bars), myriocin-treated control cells (light gray bars), mock-treated *VPS53 KD* cells (dark gray bars), or myriocin-treated *VPS53 KD* cells (black bars) is shown. Levels are plotted as fold change from that of mock-treated control cells. Error bars represent the average of three independent experiments. *p < 0.05; n.s. not significant. (**E**) Lipidomic analysis of mock-treated control fibroblasts (white bars), myriocin-treated control fibroblasts (light gray bars), mock-treated *PCCA2* patient fibroblasts (dark gray bars), or myriocin-treated *PCCA2* patient fibroblasts (black bars) is shown. Levels are plotted as fold change from mock-treated control fibroblasts. Error bars represent the average of six independent experiments. *p < 0.05; n.s. not significant.

The following figure supplements are available for figure 6:

**Figure supplement 1**. Myriocin treatment reduced accumulations of early sphingolipid intermediates due to GARP complex deficiency.

**Figure supplement 2**. LAMP1 expression in VPS53 knock-down HeLa cells is not altered.

---

Consistent with our findings on sterol accumulation in GARP mutants, previous screens for lipid droplet phenotypes have identified mutations of several GARP subunit genes (*VPS51*, *VPS53*, and *VPS54*) to be associated with neutral lipid accumulation in LDs (*Szymanski et al., 2007*; *Fei et al., 2008*), suggesting that GARP deficiency could modulate sterol metabolism. In addition, mutations in the zebrafish homolog of Vps51, *fat-free*, result in an increased number of lipid droplets in the liver and intestine (*Liu et al., 2010*).

Mutations in one component of the GARP complex, VPS53, cause PCCA2, a severe neurodegenerative disease characterized by profound mental retardation, progressive microcephaly, spasticity, and early-onset epilepsy (*Feinstein et al., 2014*). The autosomal disease, linked thus far to two PCCA2 alleles, the missense Q695R mutation, and a splice donor mutation, occurs predominantly in Jewish people of Moroccan ancestry. Our data show that one of these mutations, *VPS53 Q695R*, leads to defects in yeast that are similar to, albeit not as severe as, a Vps53 deletion. In addition, we show that sphingolipid intermediate levels are elevated in PCCA2 patient fibroblasts, which can be remedied by sphingolipid biosynthesis inhibition with myriocin. Thus, the pathogenesis of PCCA2 could be, at least partially, due to defects in lipid balance and storage.

Interestingly, the disruption of the GARP subunit *Vps54* in mice leads to a 'wobbler' phenotype, indicative of neurodegenerative disease sharing characteristics of ALS, including progressive motor degeneration and motor neuron loss (*Perez-Victoria et al., 2010*; *Moser et al., 2013*). Analyses of brain lipid profiles suggest sphingolipid intermediates and sterol esters accumulate in ALS patients (*Cutler et al., 2002*). In addition, mutations in the endo-lysosomal trafficking machinery have been implicated in neurodegenerative disorders including ALS and FTD (CHMP2B, FIG4, VAPB, and VCP) (*Yang et al., 2001*; *Parkinson et al., 2006*; *Johnson et al., 2010*; *Maruyama et al., 2010*) and defects in retrograde trafficking increase the risk for Parkinson's disease (VPS35) (*Zimprich et al., 2011*). Our data therefore suggest that lipid imbalance and toxicity due to impaired endo-lysosomal trafficking could be a common feature of this group of neurodegenerative diseases. Since inhibition of SPT reversed some aspects of cellular dysfunction in our studies, our findings suggest that inhibition of the initial step of sphingolipid synthesis may provide a useful therapeutic strategy to pursue in these diseases.

## Materials and methods

### Experimental procedures

#### Yeast strains and plasmids

All yeast strains used in this study are listed in *Supplementary file 1*. Standard yeast manipulations, including transformation, homologous recombination of PCR-generated fragments, and tetrad dissections, were performed as described previously (*Berchtold and Walther, 2009*).

All plasmids used in this study were generated using methods previously described and are listed in *Supplementary file 2*.

## Yeast culture and drug treatment

Yeast strains were grown according to standard procedures. Myriocin was added to liquid cultures at a final concentration of 1 µM. For spotting assays, myriocin, aureobasidin A, fumonisin B1, or PHS was added at concentrations as indicated and the plates incubated at 30°C for 36 hr.

## Tissue culture and RNAi

HeLa cells were cultured in Dulbecco's Modified Eagle Medium supplemented with 10% fetal bovine serum (FBS) and PenStrep (Gibco, Life Technologies, Carlsbad, CA). For RNAi, cells were treated two times (on day 1 and 4 after seeding) with 50 nM siGENOME Control siRNA RISC-free small interfering RNA (siRNA; D-001220-01-05; Dharmacon, Lafayette, CO) or VPS53 ON-TARGET plus siRNA (J-017048–08) using Oligofectamine (Invitrogen, Carlsbad, CA) according to the manufacturer's protocol. On day 6, cells were treated overnight with 1 µM myriocin or dimethyl sulfoxide (DMSO) as indicated in serum-free medium and fixed in 4% paraformaldehyde (PFA) for filipin staining or immunofluorescence.

PCCA2 were described previously (*Feinstein et al., 2014*). PCCA2 patient and control fibroblasts were cultured in Dulbecco's Modified Eagle Medium (DMEM) supplemented with 10% FBS and 1% PenStrep. Three independent human fibroblast lines were used as wild-type controls following genomic DNA sequence analysis to ensure the absence of PCCA2 mutations in *VPS53*. After reaching 80% confluence (day 3 after seeding), cells were treated with 1 µM myriocin or mock treated for 72 hr, lysed, and processed for lipidomics. Human fibroblasts were treated longer than HeLa cells to compensate for their slower growth and metabolism.

## Chemical genomics screen

The yeast deletion collection (*Winzeler et al., 1999*) and the yeast DAMP-collection (*Breslow et al., 2008*) were spotted in triplicates on SC complete plates containing 1 µM myriocin or methanol and incubated for 48 hr at 30°C. Pictures were taken with a Nikon D60 camera (Nikon, Japan) in a light box. The colony size was then measured with the program colony grid analyzer (http://sourceforge.net/projects/ht-col-measurer). The T-score was calculated as described before (*Collins et al., 2006*). The analysis and plots were done with the open source software package R (http://www.r-project.org/).

## Genetic interaction data

Data sets for the analysis of E-MAP data were derived from (*Hoppins et al., 2011*).

## Proteomics

For SILAC labeling, lysine auxotroph strains were grown in yeast nitrogen base (YNB) medium containing either 30 mg/l L-lysine or 30 mg/l L-lysine-U-(*Montefusco et al., 2014*)C$_6$,$^{15}$N$_2$. 25 optical density (OD) units of light and heavy-labeled cells were mixed and lysed in 200 µl buffer containing 50 mM Tris Cl pH = 9.0, 5% sodium dodecyl sulfate (SDS), and 100 mM dithiothreitol (DTT) for 30 min at 55°C. Lysates were cleared by centrifugation at 17000 g for 10 min and supernatants were diluted with buffer (8 M urea, 0.1 M Tris Cl pH = 8.5) to a final concentration of 0.5% SDS. Proteins were digested with the endoproteinase LysC following the protocol for filter-aided sample preparation, (*Wisniewski et al., 2009*) as described previously(*Frohlich et al., 2013*). Peptides were separated by reversed phase chromatography using 50-cm columns as described previously (*Frohlich et al., 2013*). Mass spectrometry and data analysis were performed as described previously (*Frohlich et al., 2013*).

## Lipidomics

Yeast lipids were analyzed as described previously (*Ejsing et al., 2009*; *Klose et al., 2012*; *Olson et al., 2015*). For LC-MS/MS analysis, sphingolipids were extracted from yeast according to 7.5 µg of protein by dichloromethane/methanol extraction (*Sullards et al., 2011*). The levels of PHS and DHS were determined as described previously (*Muir et al., 2014*).

Mammalian sphingolipids were analyzed as described previously (*Bird et al., 2011bib6*). For LC-MS/MS analysis, sphingolipids were extracted from HeLa cells according to 7.5 µg of protein by dichloromethane/methanol extraction (*Sullards et al., 2011*). Mammalian sphingolipids peaks were identified using the Lipid Search algorithm (MKI, Tokyo, Japan). Peaks were defined through raw files, product ion, and precursor ion accurate masses. Candidate sphingolipids and sphingolipid intermediates were identified by database (>1,000,000 entries) search of positive ion adducts. The accurate mass

extracted ion chromatograms were integrated for each identified lipid precursor and peak areas obtained for quantitation. Internal standards for each lipid class (LCB 17:0;2; Cer 18:1;2/17:0;0; HexCer 18:1;2/12:0;0; SM 18:1;2/17:0;0) spiked in prior to extraction were used for normalization and calculation of the amounts of lipids in pmol/µg protein.

## Microscopy

For fluorescence microscopy, yeast cells were grown to OD = 0.6 in synthetic medium at 30°C unless otherwise indicated. Cells were mounted in synthetic media onto coverslips previously coated with concanavalin A and imaged with a spinning-disk confocal microscope (TiLL iMIC CSU22; Andor, Northern Ireland) using a back-illuminated EM charge-coupled device camera (iXonEM 897; Andor) and a 100 × 1.4 NA oil immersion objective (Olympus, Japan). 16-bit images were collected using Image iQ (version 1.9; Andor). Images were filtered with a smoothening filter averaging 2 pixels, converted to 8-bit images, and cropped using ImageJ software (http://rsbweb.nih.gov/ij/).

## Filipin staining

Yeast cells from cultures grown to $OD_{600} \approx 0.5$ were fixed with 4% PFA for 10 min and washed three times with $H_2O$. Cells were incubated in the presence of 0.1 mg/ml filipin for 15 min in the dark at constant shaking and directly mounted on concanavalin coated coverslips. Images were taken on a Nikon epifluorescence microscope equipped with a Nikon Intensilight C-HGFIE fiber illuminator.

HeLa cells were fixed for 30 min in 4% PFA in phosphate buffered saline (PBS) and washed 3 times with PBS. Filipin (50 mg/ml) was diluted 1:250 in PBS, and cells were incubated for 30 min at room temperature in the dark, washed 3 × 5 min, and imaged directly in 6-well glass bottom plates (MatTek, Ashalnd, MA, United States). Images were collected on a DeltaVision workstation (Applied Precision, Issaquah, WA, United States) based on an inverted microscope (IX-70; Olympus) using a 100 × 1.4 NA oil immersion lens. Images were captured at 24°C with a 12-bit charge-coupled device camera (CoolSnap HQ; Photometrics, Tucson, AZ, United States) and deconvolved using an iterative-constrained algorithm and the measured point spread function (*Chi et al., 2014*).

## FM4-64 and NBD-sphingosine staining

Stock solutions of 1 mg/ml FM4-64 and 0.5 mg/ml NBD-sphingosine were prepared in DMSO. 1 OD unit of exponentially growing yeast cells were harvested by centrifugation. Cells were stained in 50 µl YPD (yeast extract peptone dextrose) medium containing 20 µg/ml FM4-64 and 10 µg/ml NBD-sphingosine for 15 min at 30°C shaking in the dark. Cells were washed 3 times with 1 ml YPD medium and incubated in 1 ml YPD for 45 min, shaking in the dark at 30°C. Cells were pelleted, resuspended in 50 µl YPD, and mounted on concavalin A coated coverslips.

## Immunofluorescence

For immunofluorescence experiments, HeLa cells were grown in 6-well glass bottom plates (MatTek) and fixed with 4% PFA/0.1 M sodium phosphate buffer pH = 7.2. Blocking and primary and secondary antibody staining were performed using 3% bovine serum albumin in PBS +0.1% Triton X-100.

## Antibodies

The primary antibodies used in this study were as follows: mouse anti-GFP antibody (Roche Life Science, Switzerland), mouse anti-HA antibody (Roche Life Science), mouse anti-PGK1 monoclonal antibody (Abcam, United Kingdom), rabbit anti-LAMP1 antibody (Cell Signaling Technology, Danvers, MA, United States), rabbit anti-Vps53 antibody (Atlas Antibodies, Sweden), and mouse anti-tubulin antibodies (Sigma, St. Louis, MO, United States). Secondary antibodies used for immunofluorescence were HRP-conjugated (horseradish peroxidase) anti-mouse and anti-rabbit IgG (Santa Cruz Biotechnology, Dallas, TX, United States) and Alexa 488–conjugated anti-rabbit IgG (Invitrogen/Thermo, Waltham, MA, United States).

## EM

High-pressure freezing and electron microscopy were performed as described previously (*Wilfling and et al., 2013*).

## Acknowledgements

We would like to thank Drs. Christopher Burd, Richard Chi, Karin Reinisch, and members of the Farese/Walther laboratory for critical discussion and comments on the manuscript. We thank Drs Ohad Birk, Juan Bonifacino, and Taroh Kinsohita for providing us with crucial reagents. We thank

Dr. Michelle Pflumm for expert assistance with manuscript preparation. We are grateful to Dr. Ole N Jensen for access to the Triversa NanoMate. This work was supported by NIGMS grant R01GM095982 (to TCW), the G Harold and Leila Y Mathers Foundation (to TCW), the Consortium for Frontotemporal Dementia Research (to RVF), and by Lundbeckfonden (R54-A5858, to CSE) and Villum Fonden (VKR023439 to CSE).

## Additional information

### Funding

| Funder | Grant reference | Author |
|---|---|---|
| National Institute of General Medical Sciences (NIGMS) | R01GM095982 | Tobias C Walther |
| G Harold and Leila Y. Mathers Foundation | | Tobias C Walther |
| University of California, San Francisco (UCSF) | Consortium of Frontotemporal Dementia Research | Robert V Farese |
| Lundbeckfonden | R54-A5858 | Christer S Ejsing |
| Villum Fonden | VKR023439 | Christer S Ejsing |

The funders had no role in study design, data collection and interpretation, or the decision to submit the work for publication.

### Author contributions

FF, Conception and design, Acquisition of data, Analysis and interpretation of data, Drafting or revising the article; CP, NK, RC, H-KH-B, MG, XL, CSE, Acquisition of data, Analysis and interpretation of data; RVF, TCW, Conception and design, Analysis and interpretation of data, Drafting or revising the article

## Additional files

### Supplementary files

• Supplementary file 1. List of all yeast strains used in this study.

• Supplementary file 2. List of all plasmids used in this study.

• Supplementary file 3. List of all hits identified in the chemical genomic screen.

• Supplementary file 4. Phenotypes of different trafficking complexes identified in the chemical genetic screen.

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
