## [Decision Letter]

Thank you for submitting your work entitled “The GARP Complex Is Required for Cellular Sphingolipid Homeostasis” for peer review at *eLife*. Your submission has been favorably evaluated by Vivek Malhotra (Senior Editor) and three reviewers, one of whom, Peter Tontonoz, is a member of our Board of Reviewing Editors. One of three reviewers, Suzanne Pfeffer, has agreed to share her identity.

The reviewers have discussed the reviews with one another and the Reviewing editor has drafted this decision to help you prepare a revised submission.

This very interesting paper shows that GARP-complex mediated retrograde transport is needed for sphingolipid homeostasis in both yeast and human cells. The authors use genome wide proteomics and lipidomics to show that loss of GARP leads to the accumulation of long chain base sphingolipid synthesis intermediates that may trigger toxicity by release of zinc from the vacuole. Ergosterol esters also increased, as did lipid droplets. Finally, a disease causing allele showed similar defects. These data reveal a surprising result: inhibition of the first step in sphingomyelin synthesis may be able to suppress neurological defects in the GARP complex.

The reviewers were in agreement that the work was of high quality and would be of interest to the *eLife* audience. All of the reviewers believed that implicating the GARP complex in sphingolipid homeostasis was an important advance.

A few straightforward issues should be addressed.

1) Does mutation of another GARP subunit show the same effect? E.g., *VPS54* mutation leads to an ALS-like phenotype in mice. Could the authors analyze sphingolipid content and myriocin-response of *VPS54* mutant cells? This would clarify whether the phenotype is a unique feature of one subunit or an entire pathway that is involved.

2) The authors stress the therapeutic implications, but the relevance of myriocin treatment for neurological disease is not yet clear, as these studies were performed in yeast and knockdown HeLa cells. The impact could be improved with a better PCCA2 model. At a minimum, analysis of the human *vps53Q624R* mutation should be performed in a human cell line.

3) In Figure 4, it is important to show whether myriocin can rescue the sterol mislocalization as in Figure 6, because this would indicate that the primary defect is with sphingolipids. Conversely, does simvastatin also partially rescue the GARP mutant sphingolipid phenotype?

4) Why do the authors think that no Ypt proteins or Imh1 showed up in the screen? Are there any GARP interactors? Please comment more on the breadth of hits beyond Figure 1—figure supplement 1.

5) The authors have not yet provided an explanation for the mis-localization of sphingolipids and the 8-fold increase in LCBs. GARP complex mediates membrane tethering, and it is not clear whether the changes in sphingolipids result directly from a retrograde trafficking defect. Do sphingolipids recycle through the retrograde vesicles controlled by retromer/GARP? Is it possible that GARP malfunction causes mislocalization of key lipid sorting proteins, which then causes mislocalization of sphingolipids? While experiments to address these questions may be beyond the scope of the present study, additional discussion of these issues could be included.

---

## [Author Response]

*1) Does mutation of another GARP subunit show the same effect? E.g.,* VPS54 *mutation leads to an ALS-like phenotype in mice. Could the authors analyze sphingolipid content and myriocin-response of* VPS54 *mutant cells? This would clarify whether the phenotype is a unique feature of one subunit or an entire pathway that is involved*.

We agree that is important to show that the different subunits of the GARP complex have similar phenotypes. We have now added the subunits *Vps52* and *Vps54* to our yeast analyses and added the data to Figure 2 and modified the text accordingly.

*2) The authors stress the therapeutic implications, but the relevance of myriocin treatment for neurological disease is not yet clear, as these studies were performed in yeast and knockdown HeLa cells. The impact could be improved with a better PCCA2 model. At a minimum, analysis of the human* vps53Q624R *mutation should be performed in a human cell line.*

The referees raise an important point. We have now obtained fibroblasts from PCCA2 patients (published earlier in Feinstein et al., J. Med. Genetics, 2014). Importantly, we observe an accumulation of early sphingolipid intermediates in these cells compared with controls. Similar findings were obtained previously in HeLa cells, where, not unexpectedly, the overall sphingolipid composition is somewhat different from skin fibroblasts. We have added these data to Figure 6 and modified the text accordingly.

*3) In*
Figure 4*, it is important to show whether myriocin can rescue the sterol mislocalization as in*
Figure 6*, because this would indicate that the primary defect is with sphingolipids*. *Conversely, does simvastatin also partially rescue the GARP mutant sphingolipid phenotype?*

We thank the reviewers for raising two interesting points. Concerning the first point, we have now analyzed sterol accumulation in *vps53* and *vps54* mutants in response to myriocin. Indeed, sphingolipid depletion is sufficient to rescue the observed accumulation of filipin-positive structures. Interestingly, myriocin treatment in wildtype cells has the opposite effect, resulting in the accumulation of filipin-positive structures. Similar effects have been observed in a mutant carrying a temperature sensitive allele of the serine palmitoyl transferase, *lcb1-100* (Baumann et al., Biochemistry, 2005). While the underlying mechanisms for this redistribution are not clear, the authors speculate that depletion of sphingolipids affects the plasma membrane ratios of sphingolipids and sterols resulting in more “free” sterols not in complex with sphingolipids, and that these “free” sterols redistribute to intracellular compartments. In line with this model, we also observe lower levels of complex sphingolipids in GARP mutants. We have added the data to Figure 3 and modified the text and Discussion accordingly.

We also agree that analysis of sphingolipids in GARP mutants treated with sterol-synthesis inhibitors would be very interesting. Unfortunately, this has proved to be very difficult due to technical reasons: growth of GARP mutants is very sensitive to inhibition of sterol biosynthesis (e.g., shown by the synthetic lethality of *vps52∆erg3∆,*
Figure 4—figure supplement 1). Therefore we were not able to achieve treatment conditions of GARP mutants with simvastatin allowing for lipid analysis.

*4) Why do the authors think that no Ypt proteins or Imh1 showed up in the screen? Are there any GARP interactors? Please comment more on the breadth of hits beyond*
Figure 1—figure supplement 1.

Thank you for raising this important point on GARP biology. Generally, we have set up our screen with stringent selection and significance criteria. Thus, it is possible that mutants with relatively smaller effects will be false negatives. We have edited the text to better describe the screen and its results.

The GARP complex is known to cooperate with the SNARE proteins Tlg1, Tlg2 and Vti1. Both *TLG2* and *Vti1*_*DAMP*_ are also strong suppressors of the growth defect caused by sphingolipid inhibition. *TLG1*_*DAMP*_ does not give a significant phenotype, which might be explained by the nature of the DAMP allele, which destabilize the mRNA of a gene (thus might not make the protein rate limiting).

YPT6 is not a significant outlier in our screen but the 98% overlapping ORF VPS63 is significantly suppressing the growth defect caused by myriocin. Possibly this could hint to a problem in the *ypt6*Δ strain in the collection. Consistent with a role for Ypt6 in maintaining lipid homeostasis, one of the two subunits of the Ypt6 nucleotide exchange factor, Ric1, is significantly more resistant to myriocin treatment.

For Imh1, we observe no defect. Thus, Imh1 could either be dispensable for GARP function in lipid homeostasis or it could be a false negative in our genetic screen.

*5) The authors have not yet provided an explanation for the mis-localization of sphingolipids and the 8-fold increase in LCBs. GARP complex mediates membrane tethering, and it is not clear whether the changes in sphingolipids result directly from a retrograde trafficking defect. Do sphingolipids recycle through the retrograde vesicles controlled by retromer/GARP? Is it possible that GARP malfunction causes mislocalization of key lipid sorting proteins, which then causes mislocalization of sphingolipids? While experiments to address these questions may be beyond the scope of the present study, additional discussion of these issues could be included*.

We thank the reviewers for pinpointing an important mechanistic question. Unfortunately, there are few good tools to study the trafficking routes of lipids in cells. Thus, the best evidence for sphingolipid accumulation due to mis-sorting of the lipids, rather than an enzyme in the biosynthetic pathway, is indirect: i) the enzymes of LCB synthesis are ER localized and we do not observe changes for their abundance or localization; ii) since long chain base levels decrease slowly after myriocin treatment in GARP mutants, we believe that sphingolipids are increasingly trafficked to vacuoles for degradation. If the alternative model was true, and the LCB increase in GARP mutants would result solemnly from de novo biosynthesis, long chain base levels should be depleted faster as SPT inhibition is near instantaneous. However, at this time, we cannot rule out that proteins important for sphingolipid synthesis and/or handling are miss-sorted in GARP mutants. We consider it an interesting model that a protein important for the transport of long chain bases out of vacuoles is not reaching its destination and therefore contributes to the observed phenotype. We have edited the Discussion to better reflect these important points.
